# Association of Pulmonary Involvement at Baseline with Exercise Intolerance and Worse Physical Functioning 8 Months Following COVID-19 Pneumonia

**DOI:** 10.3390/jcm14020475

**Published:** 2025-01-13

**Authors:** Fatma Isil Uzel, Yüksel Peker, Zeynep Atceken, Ferhan Karatas, Cetin Atasoy, Benan Caglayan

**Affiliations:** 1Department of Pulmonary Medicine, School of Medicine, Koc University, Istanbul 34010, Türkiye; uzelisil@gmail.com (F.I.U.); fkaratas@kuh.ku.edu.tr (F.K.); 2Department of Molecular and Clinical Medicine, Sahlgrenska Academy, University of Gothenburg, 40530 Gothenburg, Sweden; 3Department of Clinical Sciences, Respiratory Medicine and Allergology, Faculty of Medicine, Lund University, 22185 Lund, Sweden; 4Division of Pulmonary, Allergy, and Critical Care Medicine, University of Pittsburgh School of Medicine, Pittsburgh, PA 15213, USA; 5Department of Radiology, Koc University School of Medicine, Istanbul 34010, Türkiye; zatceken@kuh.ku.edu.tr (Z.A.); catasoy@kuh.ku.edu.tr (C.A.); 6Department of Pulmonary Medicine, Istanbul Oncology Hospital, Istanbul 34846, Türkiye; benancag@gmail.com

**Keywords:** COVID-19, radiological pulmonary involvement, exercise intolerance, health-related quality of life

## Abstract

**Objectives:** We aimed to describe the cardiopulmonary function during exercise and the health-related quality of life (HRQoL) in patients with a history of COVID-19 pneumonia, stratified by chest computed tomography (CT) findings at baseline. **Methods:** Among 77 consecutive patients with COVID-19 who were discharged from the Pulmonology Ward between March 2020 and April 2021, 28 (mean age 54.3 ± 8.6 years, 8 females) agreed to participate to the current study. The participants were analyzed in two groups based on pulmonary involvement (PI) at baseline chest CT applying a threshold of 25%. A consequent artificial intelligence (AI)-guided total opacity score (TOS) was calculated in a subgroup of 22 patients. A cardiopulmonary exercise test (CPET) was conducted on average 8.4 (±1.9) months after discharge from the hospital. HRQoL was defined using the short-form (SF-36) questionnaire. The primary outcome was exercise intolerance that was defined as a peak oxygen uptake (V′O_2peak_) < 80% predicted. Secondary outcomes were ventilatory limitation, defined as breathing reserve < 15%, circulatory limitation, defined as oxygen pulse predicted below 80%, and deconditioning, defined as exercise intolerance in the absence of ventilatory and circulatory limitations. Other secondary outcomes included the SF-36 domains. **Results:** In all, 15 patients had at least 25% PI (53.6%) at baseline chest CT. Exercise intolerance was observed in ten patients (35.7%), six due to deconditioning and four due to circulatory limitation; none had ventilatory limitation. AI-guided TOS was 30.1 ± 24.4% vs. 6.1 ± 4.8% (*p* < 0.001) at baseline, and 1.7 ± 3.0% vs. 0.2 ± 0.7% (nonsignificant) at follow-up in high and low PI groups, respectively. The physical functioning (PF) domain score of the SF-36 questionnaire was 66.3 ± 19.4 vs. 85.0 ± 13.1 in high and low PI groups, respectively (*p* = 0.007). Other SF-36 domains did not differ significantly between the groups. A high PI at baseline was inversely correlated with V′O_2peak_ (standardized β coefficient = −0.436; 95% CI −26.1; −0.7; *p* = 0.040) and with PF scores (standardized β coefficient −0.654; 95% CI −41.3; −7.6; *p* = 0.006) adjusted for age, sex, body mass index and lung diffusion capacity. **Conclusions:** One-third of participants experienced exercise intolerance eight months after COVID-19 pneumonia. A higher PI at baseline was significantly associated with exercise intolerance and PF. Notwithstanding, the radiological PI was resolved, and the exercise intolerance was mainly explained not by ventilatory limitation but by circulatory limitation and deconditioning.

## 1. Introduction

The COVID-19 pandemic caused by the new severe acute respiratory syndrome coronavirus 2 (SARS-CoV-2) resulted in considerable morbidity and mortality worldwide [1]. While most individuals infected with SARS-CoV-2 experience mild to moderate symptoms, a subset of patients develop severe pneumonia requiring hospitalization and respiratory support. Pneumonia is one of the most common complications of COVID-19, leading to acute respiratory distress syndrome (ARDS) and respiratory failure, especially in elderly people and people with complex health conditions [1].

The acute respiratory complications of COVID-19 have been extensively studied. It is also widely recognized that the long-term consequences of the disease, even in individuals who have recovered from the acute phase, can affect daily life in many aspects. Pulmonary involvement (PI) is a key determinant of disease severity in acute COVID-19 pneumonia and is commonly assessed using chest CT imaging, which reveals characteristic findings such as ground-glass opacities, consolidations, and interstitial abnormalities [2,3]. Recently, it was found that an assessment based on computer-based tomography (CT) guided by artificial intelligence (AI) helps foretell the need for oxygen and hospitalization in patients with acute COVID-19 pneumonia [4].

While the extent of PI during the acute phase of COVID-19 has been associated with disease severity and prognosis, its impact on long-term cardiopulmonary function and quality of life remains incompletely understood. Several studies from different regions have been published which investigate the effect of COVID-19 pneumonia on physical status, performance, and quality of life of patients at various time points [5,6,7,8,9].

Cardiopulmonary exercise testing (CPET) is an important tool for assessing the integrative function of cardiovascular, lung and musculoskeletal systems during exercise. The oxygen peak (V′O_2peak_) measured during CPET reflects the maximum aerobic capacity and is often reduced in patients with heart and lung disease [10].

A short-form questionnaire (SF-36) is widely used to assess the health-related quality of life (HRQoL) in several areas, including physical functioning (PF), limitations in the role due to physical and emotional health, physical pain, general health perception, vitality, social function, and mental health [11].

Understanding the relationship between PI severity at baseline and long-term cardiopulmonary function and quality of life is essential for guiding clinical management and rehabilitation strategies for patients recovering from COVID-19 pneumonia.

We hypothesized that patients with more severe radiological pulmonary involvement at baseline could have more limitations in exercise capacity and diminished HRQoL in the long term. Thus, we addressed the association between the severity of PI at baseline and cardiopulmonary function as well as HRQoL in patients eight months after COVID-19 pneumonia.

## 2. Materials and Methods

Study design and participants: This prospective cohort study included patients who were discharged from the Pulmonary Ward with a confirmed diagnosis of COVID-19 pneumonia between March 2020 and April 2021. In all, 77 patients were asked to participate in the study 6 months after discharge from hospital, and 28 of them who provided informed consent were enrolled (Figure 1). The majority (*n* = 20) of them were hospitalized during the first outbreak of the pandemic (March–May 2020). Inclusion criteria were as follows: adult patients (≥18 years) with confirmed COVID-19 pneumonia based on RT-PCR, clinical and radiological findings during hospitalization, discharged home after recovery, and willing to participate in follow-up assessments. Exclusion criteria were the presence of pre-existing cardiopulmonary diseases, musculoskeletal disorders affecting exercise capacity, and inability to perform CPET. The main reasons for not participating were unwillingness to participate, remote location to hospital, old age, severe comorbidities.

### 2.1. Data Collection

Baseline demographic and clinical data, including age, sex, body mass index (BMI), smoking history, comorbidities, duration of hospitalization, oxygen therapy during hospitalization, and pharmacological treatment, were collected from electronic medical records.

### 2.2. Pulmonary Involvement Assessment

PI severity was assessed using baseline chest CT scans obtained during hospital stay for COVID-19 pneumonia. All patients were examined with a 64-detector row CT scanner (Somatom^®^ Definition AS; Siemens Healthineers, Forchheim, Germany). Scanning was carried out in supine position and after full inspiratory breath-hold. Ground-glass opacities (GGO), consolidations, and interstitial abnormalities were recorded and summed up regarding the occupied percentage of both lungs. It was repeated when patients returned for follow-up to have a detailed evaluation at the same time as CPET. Patients were categorized into high and low PI groups based on a threshold of 25%. Chest CT images were further analyzed using artificial intelligence (Syngo Via Version VB60A_HF08) to quantify the degree of lung involvement [4] in a subgroup of 22 patients. Six patients had CT scans performed in other hospitals before admission to the study hospital and therefore could not be analyzed with the AI software. The image analysis for the pneumonia severity score was performed by an automated lung opacity analysis program, “CT Pulmonary Density”, supplied by Siemens Healthineers (Forchheim, Germany), which was previously validated externally [12]. Utilizing 3D segmentations of lesions, lungs, and lobes, the algorithm assesses the volumes of lobes and lungs along with areas of high opacity, which include ground glass and consolidation. The extent of involvement (total opacity score) was determined by calculating the ratio of the volume of high-opacity regions to the total lung volume. Two examples are illustrated in Figure 2.

### 2.3. Cardiopulmonary Function Assessment

All participants underwent CPET (Type Vyntus CPX, Viasprint 150 P, CareFusion Germany, Wurmlingen, Germany) on average 8.4 ± 1.9 months after hospital discharge. The CPET was performed using a standardized protocol on a cycle ergometer with continuous gas exchange measurement [10]. Patients were instructed to exercise to maximal exertion while respiratory gases were analyzed to determine oxygen uptake, carbon dioxide production, and ventilation. V′O_2peak_, defined as the highest oxygen uptake achieved during exercise, was expressed as a percentage of predicted values based on age, sex, and BMI. Perceived exertion and dyspnea were evaluated using the 10-point Borg scale [10].

Exercise intolerance was defined as a peak oxygen uptake (V′O_2peak_) < 80% predicted.

The cause of exercise intolerance was established for all participants with V′O_2_ peak <80% predicted. Ventilatory limitation to exercise was considered when breathing reserve was <15%. Breathing reserve was calculated as (1 − V′Epeak/maximal voluntary ventilation (MVV)) × 100%, using an estimate of forced expiratory volume in 1 s (FEV_1_) × 40 for MVV. The anaerobic threshold was estimated by the V-slope method [10]. Circulatory limitation was defined as the oxygen pulse predicted below 80% [8,10]. Oxygen pulse is the ratio of oxygen consumption (VO_2_) to heart rate (HR) (VO_2_/HR) and shows increases in stroke volume and oxygen extraction. It is considered as one of the important variables of myocardial ischemia. Deconditioning was defined as exercise intolerance when both circulatory and ventilatory limitations were eliminated.

### 2.4. Pulmonary Function Tests

Pulmonary function tests comprised measurement of spirometry and diffusing capacity of the lungs for carbon monoxide (Jaeger, Master Body PFT, CareFusion, Germany). They were applied according to ERS/ATS guidelines. Patients received detailed information and guidance about maneuvers and after at least three spirograms, the best outcomes fulfilling the criteria for repeatability and acceptability were included in the study [13,14].

A 6 min walk test (6MWT) was performed according to ATS/ERS guidelines [15] under the supervision of a nurse along the outpatient clinic corridor at the patients’ pace. Before and after walking, the respiratory and heart rate per minute, arterial blood pressure were checked, and the walking distance of each patient within six minutes was recorded.

### 2.5. HRQoL Assessment

HRQoL was assessed using the short form 36-point questionnaire (SF-36), a validated instrument comprising eight fields: physical functioning (PF), role limitations due to physical health (RP), bodily pain (BP), general health perceptions (GH), vitality (VT), social functioning (SF), role limitations due to emotional health (RE), and mental health (MH). Scores for each domain range from 0 to 100, with higher scores indicating better HRQoL.

### 2.6. Statistical Analysis

For the statistical analysis, IBM SPSS Statistics for Windows, Version 28.0 (Armonk, NY, USA: IBM Corp.) was used. Continuous variables are expressed as means ± standard deviations (SD) or medians with interquartile ranges (IQR), while categorical variables are presented as frequencies and percentages. Group differences were assessed using independent *t*-tests or Mann–Whitney U tests for continuous variables and chi-square tests for categorical variables. Linear regression analyses were conducted to determine the independent associations between PI severity and cardiopulmonary function, as well as HRQoL, adjusting for potential confounding factors such as age, sex, BMI, lung diffusion capacity, and relevant clinical variables.

## 3. Results

As shown in Figure 1, out of 28 patients (mean age 54.3 ± 8.6 years, 8 females) included in the study, 15 (53.6%) had high PI (≥25% at baseline) and 13 (46.4%) had low PI (<25%) according to the chest CT scan. Artificial intelligence (AI)-guided total opacity score (TOS) was calculated in 22 patients at baseline and before the CPETs. TOS was 30.1 ± 24.4% vs. 6.1 ± 4.8% (*p* < 0.001) at baseline, and 1.7 ± 3.0% vs. 0.2 ± 0.7% (nonsignificant) at follow-up in high and low PI groups, respectively.

As shown in Table 1, participants in the high PI group were significantly younger than in the low PI group, with no significant differences in other baseline demographic and clinical characteristics.

### 3.1. CPET

In the entire cohort, the mean V′O_2peak_ was 24.7 ± 6.3 mL/kg/min, representing 84.9 ± 17.1% of predicted values for the entire cohort (Table 2).

In all, 10 patients had exercise intolerance, of whom 6 due to deconditioning and 4 due to circulatory limitations. Two of them had hypertension but no known ischemic heart disease at baseline. None of them showed ventilatory limitation. The patients with high PI at baseline demonstrated significantly lower V′O_2peak_ compared to those with low PI (mean V′O_2peak_: 80.6 ± 15.2% vs. 97.9 ± 10.4%, *p* = 0.002). Additionally, a higher proportion of patients (*n* = 9) in the high PI group had V′O_2peak_ < 80% predicted compared to the low PI group (*n* = 1) (60.0% vs. 7.7%, *p* = 0.006) (Table 2).

### 3.2. PFT

In the entire cohort, the mean ± SD FEV_1_ was 108 ± 13.6% predicted, forced vital capacity (FVC) was 107.96 ± 13.76% predicted and DLCO was 86 ± 15.23% predicted. As shown in Table 2, only four patients (14%) had a diffusing capacity below 76% predicted. Nine out of 28 patients (32%) had a FEV_1_/FVC ratio below 80% depicting obstructive ventilatory defect. There were no restrictive defects in our cohort.

### 3.3. HRQoL

As illustrated in Table 3, the high PI group had significantly lower scores in the PF domain compared to those in the low PI group. Other domains did not differ significantly.

### 3.4. Variables Associated with Exercise Intolerance and Low PF

Multivariate regression analysis revealed that high PI at baseline was independently associated with reduced V′O_2peak_ (β = −13.39, *p* = 0.040) after adjusting for lung diffusion capacity, age, sex, BMI, oxygen therapy during hospitalization, and pharmacological treatment (Table 4). Lung diffusion capacity was also positively associated with V′O_2peak_ (β = 0.406, *p* = 0.032). AI-guided TOS at baseline of the available 22 patients showed no association with exercise tolerance and quality of life eight months after COVID-19 pneumonia.

As shown in Table 5, multivariate regression analysis revealed that high PI at baseline was independently associated with PF scores (β = −0.654, 95% CI −41.3 to −7.6, *p* = 0.006) after adjusting for age, gender, BMI, lung diffusion capacity, V’O_2peak_ and pulmonary involvement. However, there were no significant differences in other SF-36 domains between the two groups. While male gender showed a trend towards a negative association with PF scores (β = −0.393, *p* = 0.084), this association did not reach statistical significance.

## 4. Discussion

The main finding of the current study was that one-third of participants experienced exercise intolerance eight months after COVID-19 pneumonia. Moreover, higher severity of PI at baseline, as assessed by chest CT imaging, was associated with impaired cardiopulmonary function, as evidenced by reduced V′O_2peak_, and poorer physical functioning, as indicated by lower PF scores on the SF-36 questionnaire. The exercise intolerance was mainly explained by the circulatory limitation and deconditioning. The fact that there was no ventilatory limitation in the subgroup of patients with exercise intolerance, but deconditioning was the limiting factor in 60% of the patients, raises the suspicion that the impact of the disease was not via the ventilatory function.

Previously, Ingul CB et al. [5] investigated the cardiopulmonary function of COVID-19 patients three and 12 months post-discharge in their prospective, longitudinal, multicenter cohort study. Two hundred and ten patients with valid CPET at either three or 12 months post-discharge were included in the analyses. They did not evaluate the pulmonary involvement and quality-of-life aspects as we did. It was concluded that 1 year after hospital discharge, the majority (77%) of the patients had no exercise intolerance. Deconditioning in their cohort was also the leading exercise-limiting factor.

Another study conducted in the first phase of the pandemic investigated exercise capacity via CPET 3–4 months after hospital discharge [6]. Multifactorial exercise-limiting factors were observed in one third (31%) of the patients, where exercise intolerance was defined as V′O_2_ peak < 80% predicted. Deconditioning was the major cause in 63% of the cohort. They did not include pulmonary involvement and quality-of-life evaluation, but self-reported dyspneic patients had significantly lower V′O_2_ peak.

Lerum TG et al. [7] aimed to describe dyspnea, quality of life, lung functions and chest CT findings 3 months after discharge by comparing patients with and without Intensive Care Unit (ICU) admission. Most of the patients had lung function values within normal limits regardless of ICU admission status, as we observed in our cohort, but approximately half of the patients reported persistent dyspnea. Pathological CT findings were significantly higher in patients admitted to an ICU, but quality-of-life scores were not different between the two groups. Only a greater number of participants who were admitted to the ICU indicated a reduced capacity to carry out their typical activities. Likewise, we noted a relationship with baseline pulmonary involvement and physical functioning domain of SF-36 in our cohort 8 months after discharge.

In a similar prospective cohort study, but with paired historical controls, da Silveira et al. [8] assessed 47 COVID-19 patients who had been previously hospitalized and compared them with 141 matched controls. The median duration from hospital discharge to cardiopulmonary exercise testing (CPET) was 7 months. Among the COVID-19 patients, 45% experienced exercise intolerance, whereas only 8.5% of the healthy controls did. None of them had ventilatory limitation, but the majority (57%) had cardiocirculatory limitation in contrast to our findings. They found an association of the presence of coronary artery disease with a lower predicted-percentage peak VO_2_ (ppVO_2_). In the literature, it has been suggested that exercise intolerance could also be influenced by the COVID-19-induced cardiac alterations, which are often encountered in patients with lung injury on chest computed tomography [16]. In our study, only four patients exhibited circulatory limitation, and two of them had hypertension but no known ischemic heart disease at baseline. Thus, we cannot exclude a possible COVID-19-induced circulatory limitation in those patients.

Regarding quality-of-life assessment with SF-36, it has been suggested that both physical and mental mean scores were diminished in COVID-19 patients [8]. In that study, the physical functioning domain did not show a statistically significant difference between patients and controls, although it was found to be reduced in the patient group. These findings support our results suggesting ongoing reduced functional capacity and HRQoL of the patients compared to matched controls even after more than 6 months post-discharge. Severe disease defined by clinical signs of pneumonia was also associated with reduced ppVO_2_. This was in accordance with our results relating increased baseline pulmonary involvement with reduced exercise tolerance months later.

In another study, Rinaldo RF et al. included patients with low and normal V’O_2_ at 3–6 months to repeat CPET at a mean time from hospital discharge of 24 months [9]. No patient had received a structured rehabilitation program after discharge. In all, 65% of the V’O_2_-low patients had fully recovered while the main final reason for exercise intolerance was deconditioning. This conclusion supports our findings in that deconditioning is the most important aspect of exercise intolerance even in the long term.

Atçeken et al. [4] assessed the relationship between AI-assisted CT-derived severity scores (SSs) and the short-term outcomes of COVID-19 patients during the initial outbreak of the pandemic. Their findings showed that AI-assisted CT-derived severity scores can be utilized to anticipate the requirement for supplemental oxygen and hospital admission in individuals with COVID-19 pneumonia. We were unable to reveal an association between the AI-guided CT-based total opacity score and long-term exercise intolerance or the HRQoL of the patients. This may be explained by the limited number of chest CTs we could evaluate via AI-guidance.

Finally, in their prospective, longitudinal study enrolling hospitalized moderate–severe COVID-19 patients, Boschino M. et al. [17] performed a chest CT 12 months after discharge. They revealed that, after one year, no honeycombing was present and 93% of participants had complete resolution of baseline lung abnormalities. This relieving result is compatible with our results that the radiological findings nearly completely resolved after more than 8 months.

The observed association between baseline PI severity and reduced V′O_2peak_ is consistent with previous studies demonstrating a link between the extent of lung involvement and persistent respiratory dysfunction following COVID-19 pneumonia.

The impact of PI on physical functioning and quality of life is also notable, with patients in the high PI group reporting lower PF scores compared to those with low PI. Physical functioning encompasses the ability to perform activities of daily living and engage in physical activities, and its impairment can significantly affect overall well-being and functional independence. The observed association between PI severity and reduced PF scores underscores the importance of early identification and targeted interventions to mitigate the long-term consequences of COVID-19 pneumonia on physical function and quality of life. As also highlighted in an umbrella review [18], physical activity relates to overall health outcomes, which is particularly relevant given the deconditioning observed among our participants. It has also been demonstrated that physical activity is associated with reduced mortality and disease outcomes in COVID-19 [19].

The mechanisms underlying the association between PI severity and long-term outcomes in COVID-19 pneumonia are multifactorial and likely involve a combination of microscopic pulmonary fibrosis undetectable with standard imaging modalities, respiratory muscle weakness, cardiovascular dysfunction, and systemic inflammation.

The primary limitations of our study include the small sample size of participating patients from a single institution and the lack of a control group for comparing the radiological findings and quality-of-life outcomes. However, this study was a longitudinal follow-up study to address cardiopulmonary function in patients 8 months after suffering COVID-19 pneumonia, and we chose to compare the subgroups with high versus low pulmonary involvement on CT at baseline. We acknowledge that it would be the best way to follow the natural course of the cardiopulmonary function and exercise intolerance at baseline and 8 months after the COVID-19 onset; however, it was not possible to perform lung function tests and CPET during the acute phase of COVID-19 pneumonia. Moreover, there was no objective measurement of participants’ pre-COVID cardiopulmonary function or exercise capacity, making it difficult to establish the true extent of functional decline attributable to COVID-19. All patients included in the study were hospitalized during the initial phase of the pandemic, which may result in different outcomes in vaccinated individuals and those who had access to antiviral treatments. Additionally, there is a possibility of unaccounted confounding factors that could impact the observed associations. Furthermore, we did not incorporate objective measures to assess the baseline functional status and exercise capacity of our study population.

## 5. Conclusions

In conclusion, this study provides novel insights into the long-term effects of PI on cardiopulmonary function and quality of life in patients recovering from COVID-19 pneumonia. Higher severity of PI at baseline was independently associated with reduced peak oxygen uptake and poorer physical functioning eight months post-discharge. These findings highlight the importance of early identification and targeted interventions for patients with significant pulmonary involvement to optimize long-term outcomes and quality of life following COVID-19 pneumonia. Future research is warranted to further elucidate the mechanisms underlying post-COVID-19 respiratory sequelae and evaluate the effectiveness of pulmonary rehabilitation interventions in this population.

## Figures and Tables

**Figure 1 jcm-14-00475-f001:**
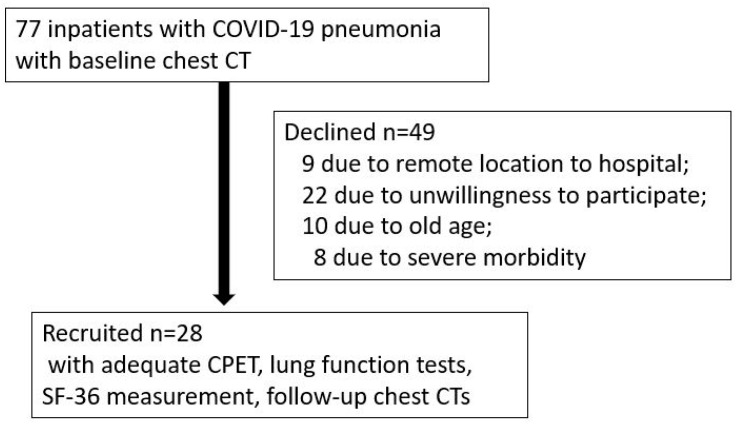
Flow chart of the study. COVID-19 = coronavirus disease; CT = computed tomography.

**Figure 2 jcm-14-00475-f002:**
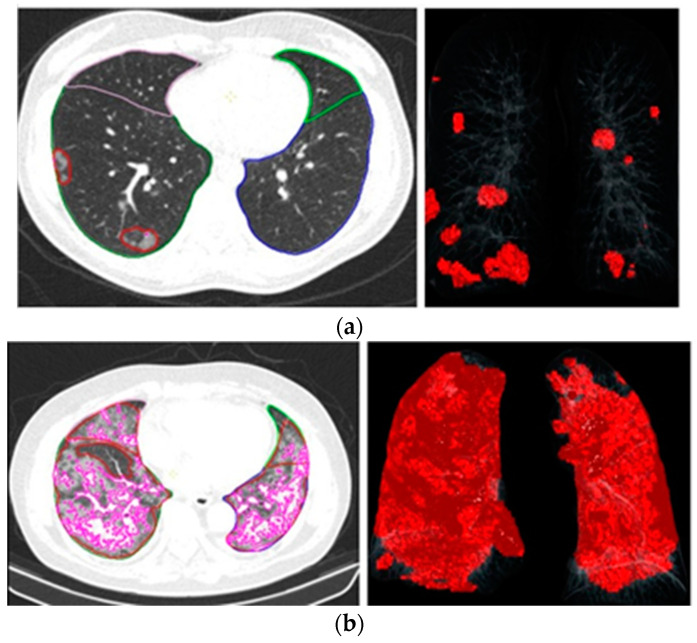
(**a**) Chest tomography images and artificial intelligence calculations of a patient with a total opacity score of 1.1; (**b**) chest tomography images and artificial intelligence calculations of a patient with a total opacity score of 18.0.

**Table 1 jcm-14-00475-t001:** Baseline characteristics of study population with relation to radiological pulmonary involvement.

Characteristics	High PI Group (*n* = 15)	Low PI Group (*n* = 13)	*p*-Value
Age, years	50.4 ± 9	58.9 ± 5.3	0.007
Female sex, *n* (%)	5 (33.3)	3 (23.1)	0.410
BMI, kg/m^2^	31.4 ± 7.5	30.5 ± 5	0.721
Smoker, *n* (%)	4 (26.7)	6 (46.2)	0.433
Hypertension, *n* (%)	6 (40.0)	5 (38.5)	0.929
In-hospital days, (median [IQR])	9 (7–11)	8 (6–10)	0.448
In-hospital oxygen therapy, *n* (%)	7 (46.7)	6 (46.2)	0.979
Corticosteroid use, *n* (%)	8 (53.3)	7 (53.8)	0.964
Antivirals, *n* (%)	9 (60.0)	8 (61.5)	0.928
Anticoagulants, *n* (%)	4 (26.7)	3 (23.1)	0.772

Abbreviations: BMI, body mass index; PI, pulmonary involvement; IQR, interquartile range.

**Table 2 jcm-14-00475-t002:** Pulmonary function test and CPET results with relation to radiological pulmonary involvement. Data are presented as mean ± standard deviation.

Characteristics	High PI Group (*n* = 15)	Low PI Group (*n* = 13)	*p*-Value
FVC (mL)	4130 ± 909	3813 ± 881	0.360
FVC (%)	104.7 ± 15	111.7 ± 11	0.187
FEV_1_ (mL)	3369 ± 759	3111 ± 761.6	0.379
FEV_1_ (%)	105.5 ± 13.9	112.3 ± 12.6	0.189
FEV_1_/FVC (%)	81.7 ± 5.4	81.4 ± 3.7	0.861
DLCO (%)	84 ± 17	89.6 ± 12.9	0.358
V’O_2peak_ (mL/kg/min)	21.5 ± 5.4	22.3 ± 4.6	0.697
V’O_2peak_ (%)	80.6 ± 15.2	97.9 ± 10.4	0.002
6 MWT (m)	465 ± 30	472.5 ± 60	0.673

Abbreviations: FVC, forced vital capacity; FEV1, forced expiratory volume in 1 s; DLCO, diffusing capacity for carbon monoxide; V’O_2peak_, peak oxygen uptake; 6 MWT, 6 min walking test.

**Table 3 jcm-14-00475-t003:** SF-36 domain scores with relation to radiological pulmonary involvement. Data are presented as mean ± standard deviation.

SF-36 Domains	High PI Group (*n* = 15)	Low PI Group (*n* = 13)	*p*-Value
Physical functioning (PF)	66.3 ± 19.4	85.0 ± 13.1	0.007
Role limitations due to physical health (RP)	83.3 ± 15.4	84.6 ± 16.3	0.832
Bodily pain (BP)	83.3 ± 12	78.5 ± 18.5	0.411
General health perceptions (GH)	65 ± 13.6	67 ± 15	0.698
Vitality (VT)	62 ± 19.6	58 ± 19.8	0.604
Social functioning (SF)	75.8 ± 25.6	75.9 ± 25	0.989
Role limitations due to emotional health (RE)	88.9 ± 24	89.8 ± 15.9	0.906
Mental health (MH)	73.7 ± 13	71 ± 15	0.620

Abbreviations: SF-36, short-form 36-item questionnaire; PI, pulmonary involvement.

**Table 4 jcm-14-00475-t004:** Univariate and multivariate regression analysis results with the dependent variable V’O_2peak_ (%).

	Univariate			Multivariate		
Variable	β-Coeff	95% CI	*p*-Value	β-Coeff	95% CI	*p*-Value
Age	0.341	−0.7, 1.318	0.076	0.172	−0.520, 0.863	0.611
Gender	0.108	−9.93, 17.28	0.583	2.286	−13.11, 17.69	0.761
BMI	0.073	−0.810, 1.168	0.712	0.289	−0.744, 1.322	0.567
DLCO	0.439	0.078, 0.821	0.020	0.406	0.039, 0.774	0.032
Pulmonary involvement	0.56	−27.480, −6.940	0.002	−13.39	−26.08, −0.701	0.040

Abbreviations: β-Coeff, beta coefficient; CI, confidence interval.

**Table 5 jcm-14-00475-t005:** Multivariate regression analysis results with the dependent variable physical functioning (PF) domain score of SF-36.

	Multivariate		
Variable	β-Coeff	95% CI	*p*-Value
Age	−0.171	−1.217, −0.457	0.356
Gender	−0.393	−34.78, 2.35	0.084
BMI	0.071	−1.04, 1.47	0.727
DLCO	0.239	−0.19, 0.79	0.221
V’O_2peak_	−0.050	−0.593, 0.471	0.815
Pulmonary involvement	−0.654	−41.3, −7.6	0.006

## Data Availability

Individual participant data reported in this article can be provided by contacting the corresponding author.

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
