# Peer review of "Association of Pulmonary Involvement at Baseline with Exercise Intolerance and Worse Physical Functioning 8 Months Following COVID-19 Pneumonia"

_jcm, 2025, doi:10.3390/jcm14020475_

Round 1
Reviewer 1 Report
Comments and Suggestions for Authors
This study focuses on the long-term cardiorespiratory recovery of patients with an acute COVID-19 infection.
I have some suggestions for the authors:
1. In the Abstract, the sentence "Notwithstanding, the radiological PI was resolved, and the exercise intolerance was mainly 38 explained by the circulatory limitation and deconditioning" is confusing and should be rephrased.
2. The authors mention only the pulmonary involvement in COVID-19, but they neglect the impact of this virus on the heart. The primary outcome of this study, exercise intolerance, could also be influenced by the COVID-19 induced cardiac alterations, which are often encountered in patients with lung injury on chest computed tomography, as mentioned in the following manuscript: Tudoran, C.; Tudoran, M.; Cut, T.G.; Lazureanu, V.E.; Bende, F.; Fofiu, R.; Enache, A.; Pescariu, S.A.; Novacescu, D. The Impact of Metabolic Syndrome and Obesity on the Evolution of Diastolic Dysfunction in Apparently Healthy Patients Suffering from Post-COVID-19 Syndrome. Biomedicines 2022, 10, 1519. https://doi.org/10.3390/biomedicines10071519.
3. The major limitation of this study is the very small sample size of 28 patients. It could affect the results and the statistical analyses. I suggest the authors to perform a sample size analysis and if possible, to increase the number of patients.
4. Since the authors re-evaluated the patients 8 month after the initial COVID-19 infection, they could present the evolution, starting from the first examination, during the acute phase and ending 8 months later.
Comments on the Quality of English LanguageEnglish Editing is neede.
Reviewer 2 Report
Comments and Suggestions for Authors
Dear authors,
I have now completed the review of the manuscript titled "Association of pulmonary involvement at baseline with exercise intolerance and worse physical functioning 8 months following COVID-19 pneumonia."
In the present study, the authors answered an important clinical question regarding the long-term effects of COVID-19 pneumonia on cardiopulmonary function and quality of life. The researchers used comprehensive assessment methods, including cardiopulmonary exercise testing, CT imaging with AI analysis, and validated quality of life measurements. The study design appropriately incorporated multiple outcome measures and used robust statistical analyses, including multivariate regression to account for potential confounding factors. The follow-up period of approximately 8 months provides valuable insights into medium-term outcomes.
However, there are some issues to be modified:
1. The absence of a control group makes it difficult to attribute the observed effects specifically to COVID-19 pneumonia versus other factors that might affect cardiopulmonary function and quality of life over time.
2. Various aspect of physical activity should be discussed more. For example, how physical activity relates to overall health outcomes, which is particularly relevant given the deconditioning observed in study's patients. I would like to recommend authors to mention Physical activity and prevention of mental health complications: an umbrella review.
3. Also, I would like to recommend authors to discuss Baseline physical activity is associated with reduced mortality and disease outcomes in COVID-19: A systematic review and meta-analysis. Since this article would complement current study by showing how pre-existing physical activity levels might influence COVID-19 outcomes, helping explain varying degrees of exercise intolerance.
4. There was no objective measurement of participants' pre-COVID cardiopulmonary function or exercise capacity, making it difficult to establish the true extent of functional decline attributable to COVID-19.
5. The artificial intelligence component of the CT analysis was only available for 22 of the 28 participants, further reducing the statistical power for this aspect of the analysis. The study doesn't adequately explain why this limitation existed or its potential impact on the findings.
Thank you for your valuable contributions to our field of research. I look forward to receiving the revised manuscript.
Round 2
Reviewer 1 Report
Comments and Suggestions for Authors
The authors answered almost all my questions.
Reviewer 2 Report
Comments and Suggestions for Authors
All comments have been thoroughly addressed. I extend my gratitude to both the authors and editors for taking my opinions into consideration during the review of this manuscript.